# Kinetics of TTV Loads in Peripheral Blood Mononuclear Cells of Early Treated Acute HIV Infections

**DOI:** 10.3390/v15091931

**Published:** 2023-09-15

**Authors:** Isabella Abbate, Gabriella Rozera, Eleonora Cimini, Fabrizio Carletti, Eleonora Tartaglia, Marika Rubino, Silvia Pittalis, Rozenn Esvan, Roberta Gagliardini, Annalisa Mondi, Valentina Mazzotta, Marta Camici, Enrico Girardi, Francesco Vaia, Vincenzo Puro, Andrea Antinori, Fabrizio Maggi

**Affiliations:** 1Laboratory of Virology, National Institute for Infectious Diseases “Lazzaro Spallanzani” IRCCS, 00149 Rome, Italy; isabella.abbate@inmi.it (I.A.); gabriella.rozera@inmi.it (G.R.); fabrizio.carletti@inmi.it (F.C.); fabrizio.maggi@inmi.it (F.M.); 2Laboratory of Cellular Immunology and Pharmacology, National Institute for Infectious Diseases “Lazzaro Spallanzani” IRCCS, 00149 Rome, Italy; eleonora.tartaglia@inmi.it (E.T.); marika.rubino@inmi.it (M.R.); 3AIDS Referral Center, National Institute for Infectious Diseases “Lazzaro Spallanzani” IRCCS, 00149 Rome, Italy; silvia.pittalis@inmi.it (S.P.); rozenn.esvan@inmi.it (R.E.); vincenzo.puro@inmi.it (V.P.); 4Clinical Infectious Department, National Institute for Infectious Diseases “Lazzaro Spallanzani” IRCCS, 00149 Rome, Italy; roberta.gagliardini@inmi.it (R.G.); annalisa.mondi@inmi.it (A.M.); valentina.mazzotta@inmi.it (V.M.); marta.camici@inmi.it (M.C.); andrea.antinori@inmi.it (A.A.); 5Scientific Direction, National Institute for Infectious Diseases “Lazzaro Spallanzani” IRCCS, 00149 Rome, Italy; enrico.girardi@inmi.it; 6General Direction, National Institute for Infectious Diseases “Lazzaro Spallanzani” IRCCS, 00149 Rome, Italy; francesco.vaia@inmi.it

**Keywords:** torquetenovirus, HIV, acute infection, T cell, immunity, senescence

## Abstract

Torquetenovirus (TTV) is the most abundant component of the human blood virome and its replication is controlled by a functioning immune system. In this study, TTV replication was evaluated in 21 people with acute HIV infection (AHI) and immune reconstitution following antiretroviral therapy (ART). PBMC-associated TTV and HIV-1 DNA, as well as plasma HIV-1 RNA, were measured by real-time PCR. CD4 and CD8 differentiation, activation, exhaustion, and senescence phenotypes were analyzed by flow cytometry. Thirteen healthy donors (HD) and twenty-eight chronically infected HIV individuals (CHI), late presenters at diagnosis, were included as control groups. TTV replication in AHI seems to be controlled by the immune system being higher than in HD and lower than in CHI. During ART, a transient increase in TTV DNA levels was associated with a significant perturbation of activation and senescence markers on CD8 T cells. TTV loads were positively correlated with the expansion of CD8 effector memory and CD57+ cells. Our results shed light on the kinetics of TTV replication in the context of HIV acute infection and confirm that the virus replication is strongly regulated by the modulation of the immune system.

## 1. Introduction

Torquetenovirus (TTV) is the oldest member of the *Anelloviridae* family (discovered in 1997) [1] and infects most of the human population by inducing a chronic active infection with no associated clinical manifestation. TTV represents the major component of the human blood virome [2,3] and it may replicate in peripheral activated T cells, but the preferential cell type supporting its replication has not yet been clearly identified [4,5,6]. Increasing evidence exists on the control of TTV replication exerted by the immune system, with TTV loads that are higher in the blood of subjects with an impaired immune status compared to healthy controls. Strong evidence arguing for the important role of the immune system in the equilibrium that TTV establishes with its hosts also includes data from HIV-1-infected individuals. Studies, in which the patients’ loads of TTV were measured, with a few exceptions [7], have produced evidence arguing for the existence of a correlation between the severity of the HIV + donors’ immunosuppression and the loads of TTV carried. One of the first studies [8] showed that 51% of people living with HIV (PLWH), who were also TTV-positive, had TTV loads greater than those detected among healthy donors, and that HIV + donors with significantly higher TTV loads had lower CD4+ T cell counts. Similarly, other studies [9,10,11] confirmed the inverse association between high TTV levels and low CD4+ T cell numbers. More recently, Schmidt et al. [12] found that the median TTV levels were significantly higher in subjects with a poor CD4+ T cell recovery (<50 cells/mm^3^) after antiretroviral therapy (ART) than in PLWH with a CD4+ T cell recovery of more than 200 cells/mm^3^ after 1 year of treatment. All these studies strongly confirmed the association between an impaired immune system, such as that of PLWH, and a significant increase in TTV replication. This association is still more evident in subjects who undergo transient changes following perturbations in immunity (e.g., transplant patients). In this setting of subjects, TTV loads have been proven to be useful to monitor patients’ immunity, predicting post-transplant clinical adverse outcomes (i.e., microbial infections and graft rejection), and guiding pharmacological interventions [13,14]. However, the association between TTV load and immune cell counts in this setting provided conflicting results, especially in the allogenic hematopoietic stem cell transplantation (HSCT) context [15,16,17], whereas a correlation between TTV loads and T-cell proliferation capacity was recently demonstrated [18].

The study aimed to investigate TTV replication in HIV acute infection (AHI) and, in concordance with the hypothesis that TTV load may be useful as an early marker of immune reconstitution, evaluate cellular-associated TTV loads after early administration of antiretroviral therapy. 

## 2. Materials and Methods

### 2.1. Study Populations

Twenty-one HIV + donors with AHI from the National Institute for Infectious Diseases (INMI) observational cohort of primary infection SIREA (SIndrome REtrovirale Acuta cohort) were evaluated at serodiagnosis (T0, n = 21), at 3 (T0.5, n = 12), and at 12 (T1, n = 17) months after receiving ART with tenofovir/emtricitabina (TDF/FTC) associated with either darunavir/ritonavir (DRV/RTV) and raltegravir (RGV) or dolutegravir (DLG). Serodiagnosis of AHI was characterized by either the combination of an HIV Ab/Ag combo positive and an immunoblot negative assay (n = 16, Fiebig II/III stage) or an HIV Ab/Ag combo positive and an immunoblot indeterminate assay (n = 5, Fiebig IV stage), according to WHO criteria for Western blot confirmatory assay, i.e., two *env* reactive bands. A total of 13 healthy donors (HD) and 28 HIV chronically HIV + donors (CHI), late presenters at diagnosis (CD4 < 200 or AIDS) and naïve to antiretroviral treatment, were also evaluated as control groups. 

### 2.2. Peripheral Lymphocyte Isolation

PBMC samples of HIV and HD subjects were isolated by gradient centrifugation (Lympholyte, Cedarlane, Burlington, ON, Canada) counted in trypan blue, and frozen in fetal bovine serum (FBS, Euroclone, Milan, Italy) with 10% dimethyl sulfoxide (DMSO, Euroclone).

### 2.3. TTV Quantification

TTV DNA was extracted from PBMC by the QIAsymphony DNA Midi Kit (QIAGEN, S.r.l. Milan, Italy), and then amplified by a quantitative in-house real-time PCR (qPCR), as described by Maggi et al. [19]. The assay amplifies a short fragment of the untranslated region of the TTV genome, which is highly conserved among all TTV species hitherto known. The following primer/probe sequences were used: forward primer GTGCCGIAGGTGAGTTTA, position 177–194, reverse primer AGCCCGGCCAGTCC, position 226–239, and probe FAM-TCAAGGGGC AATTCGGGCT-NFQ, position 205–223. qPCR was performed in a final volume of 25 μL containing 5 μL of extracted TTV DNA. Amplification was performed on the Qiagen Rotor-Gene Q. Tenfold dilutions of a plasmid containing the target region were used to build the standard curve for TTV quantification. An additional real-time PCR targeting the housekeeping cellular hTERT gene was used to refer TTV DNA copies to the number of cells present in the analyzed samples, and TTV loads were expressed as log copies per 10^6^ cells. This in-house real-time PCR displayed a LOD of 3 copies per reaction volume. 

### 2.4. HIV Quantification

Plasma HIV-1 RNA was measured by Aptima™ HIV-1 Quant Dx Assay (Hologic, Bedford, MA, USA). The assay uses specific target-capture transcription-mediated amplification (TMA), targets highly conserved regions of HIV-1 polymerase (*pol*) and *long terminal repeat* (LTR), and runs on the fully automated Panther platform. It has high sensitivity and a broad dynamic range for HIV-1 detection and quantitation, with a lower limit of quantitation of 30 copies/mL and a 95% limit of detection of 12 copies/mL. Total HIV-1 DNA was quantified by real-time PCR targeting the LTR region in the same nucleic acid extracted from PBMC used for TTV DNA quantification with the sense primer NEC 152 (GCCTCAATAAAGCTTGCCTTGA) and the reverse primer NEC 131 (GGCGCCACTGCTAGAGATTTT) in the presence of a dually (FAM and TAMRA) labeled NEC LTR probe (AAGTAGTGTGTGCCCGTCTGTTRTKTGACT). As a standard curve, dilutions of 8E5 cell DNA containing 1 proviral copy per cell were used [20]. As for TTV DNA loads in PBMC, total HIV-1 DNA was expressed as log copies/10^6^ cells using the number of cells detected in the sample volume that underwent amplification by the hTERT gene targeted in real time. Also in this case, the in-house real-time PCR used to quantify total HIV-1 DNA had a LOD of 3 copies/reaction volume. 

### 2.5. Phenotypic Staining and Flow Cytometry Analysis

The differentiation profiles, activation, exhaustion, and senescence of CD4+ and CD8+ T cells were analyzed by flow cytometry, using a dried reagent tube (DuraClone IM T cell subsets tube, Beckman Coulter, Hialeah, FL, USA). The DuraClone tube contained the following antibodies: CD45RA-FITC, CCR7-PE, CD28-ECD, PD1-PC5.5, CD27-PC7, CD4-APC, CD8-A700, CD3-APCA750, CD57-Pacific Blue, and CD45-Krome Orange. We added the anti-CD38 antibody BV610 (Beckman Coulter) to the Duraclone tube. Briefly, PBMC were defrosted in complete medium (RPMI-1640 supplemented with 10% fetal bovine serum, 2 mmol glutamine, 50 IU/mL penicillin, and 50 µg/mL streptomycin; EuroClone, Milan, Italy), added to the DuraClone tube and incubated for 15 min, at room temperature. After incubation, VersaLyse Lysing Solution (Beckman Coulter) was added and incubated for 15 min. Finally, samples were washed with PBS 1X, fixed with paraformaldehyde 1X, and then acquired by DxFLEX (Beckman Coulter). Samples were analyzed using DxFLEX software 2.0 (Beckman Coulter). The combined positivity to the pair of CD45RA/CCR7 immunity surface markers on CD4+ and CD8+ T cells was used to assess the percentages of naïve (CD45RA+ CCR7+), central memory (CM, CD45RA-CCR7+), effector memory (EM, CD45RA-CCR7−), and terminally differentiated effector memory (TEMRA, CD45RA+CCR7−) cells, as in [21].

### 2.6. Statistical Analysis

Categorical data were presented as frequencies and percentages, while continuous variables were presented as a median and interquartile range (25th percentile; 75th percentile), as appropriate. The Fisher’s exact test (FET, for categorical variables), Wilcoxon test (WT), and the Mann–Whitney test (MWT) (for intra- and inter-group differences, respectively, and for quantitative variables) were used for comparisons as appropriate. A non-parametric Spearman correlation test (SCT) was used to assess a possible linear association between two continuous variables. Prism 8.0.2 was used for univariate statistical analyses. Multivariable logistic regression analysis was performed using SPSS V.23 for Windows. Tests were two-sided, and *p* < 0.05 was considered statistically significant. 

## 3. Results

### 3.1. Viro-Immunological Features in AHI at Serodiagnosis

Table 1 summarizes the virological and immunological features of the AHI at serodiagnosis (T0). AHI were, with only one exception, all male, with a median (IQR) age of 39 (33–49) years, median (IQR) CD4+ T cells/mm^3^ of 569 (284–787), median (IQR) CD4/CD8 ratio of 0.7 (0.5–1.1), median (IQR) HIV-1 RNA log copies/mL of 7.3 (6.1–8.0), median (IQR) HIV-1 DNA log copies/10^6^ PBMC of 4.2 (3.9–4.9), and median (IQR) TTV DNA log copies/10^6^ PBMC of 3.5 (2.4–4.8). Basal levels of PBMC-associated TTV DNA in AHI were then compared to those found in HD and CHI, with the latter characterized by low CD4+ T cell counts [median (IQR): 67 (35–128) cells/mm^3^]. As shown in Figure 1A, TTV was differently prevalent among the study populations (*p* = 0.0038 AHI vs. HD and *p* = 0.07 AHI vs. CHI in FET), and TTV loads in AHI were significantly higher than in HD (*p* = 0.041) and lower than in CHI (*p* < 0.0001) in MWT. At this time, a significant positive correlation between PBMC-associated TTV DNA and plasma HIV-1 RNA (r = 0.447, *p* = 0.042; Figure 1B) and a trend towards an inverse correlation between HIV-1 RNA and CD4+ T cell counts (r = −0.384, *p* = 0.085; Figure 1C) were observed. In addition, we performed multivariate logistic regression analysis by combining AHI and CHI TTV DNA, HIV-1 RNA, and HIV-1 DNA values and categorizing patients for CD4 median value (≥ or <150 cells/mm^3^) to evaluate independent variables potentially able to influence the CD4+ T cell counts. The results indicated that TTV viral load was the only statistically significant variable independently associated with CD4 T cell counts, and that it increased in value with a decrease in CD4 T cell count, *p* = 0.003, O.R. 0.51 (95%C.I. 0.33–0.79). 

In a subgroup of 11 AHI, for whom DMSO-frozen PBMC samples were available, CD4+ and CD8+ T cell phenotypic profile (CD45RA, CCR7), markers of activation (CD38), exhaustion (PD-1), and senescence (CD57) were measured, and compared to those found in a control group of 13 HD. Multi-parametric flow cytometry was performed on both CD4+ and CD8+ subpopulations to evaluate CD45RA/CCR7 expression and to discriminate naïve, CM, EM, and TEMRA cells, as described in the methods section. Figure 2A,B shows in AHI a significant reduction in naïve cells on both CD4+ and CD8+ T cells (*p* = 0.01), an increase in EM on both CD4+ and CD8+ T cells (*p* = 0.02 and *p* = 0.0001, respectively), and a reduction in CM only on CD8+ T cells with respect to HD (*p* = 0.003). Additionally, when CD4+ and CD8+ T cell markers in AHI were compared with those found in HD, percentages of CD38+, and CD57+ cells were significantly higher in both CD4+ and CD8+ T cell subpopulations (*p* = 0.003 in CD38 CD8+; *p* = 0.0001 CD38 CD4+; *p* = 0.004 CD57 CD8+; Figure 2C,E), whereas PD-1+ cells increased in only the CD8+ T cell subset (*p* = 0.003 in PD-1 CD8+, Figure 2D). 

### 3.2. Viro-Immunological Features in 1-Year Treated AHI

Kinetics of PBMC-associated TTV DNA loads were studied during the first year of early administrated ART [median (IQR): 4 (2–7) days after serodiagnosis]. A peak of TTV DNA in PBMC was observed at 3 months post-therapy start (T0.5), while HIV-1 RNA and DNA levels significantly declined at T1 (at one year of therapy), indicating that ART was effective in limiting HIV replication in the studied HIV+ donors (Figure 3A). Interestingly, the TTV DNA increase at T0.5 was nearly inversely correlated with TTV DNA levels at T0 (r = −0.570, *p* = 0.05). During one year of ART, values of CD4+ T cells and CD4/CD8 ratio significantly increased (Figure 3B). Despite a similar trend between TTV DNA and CD4+ T cell kinetics, TTV loads at T0 were unrelated to either CD4+ T cell counts at T1 (r = 0.125, *p* = 0.63) or with the percentages of patients with a CD4+ T cell counts increment of more than 200 at T1. In fact, the percentage of patients who displayed an increment of more than 200 cells at T1 was not statistically different comparing people with a high TTV DNA (>3.5 log copies/10^6^ cells) or with a low TTV DNA at T0 (<2.5 log copies/10^6^ cells), i.e., 50 vs. 30% of patients (*p* > 0.05). When CD4+ and CD8+ differentiation profiles at T0 and on-ART were compared, no significant changes in the percentage of naïve, CM, EM, and TEMRA cells were observed in both CD4+ and CD8+ T cells, except for naïve cells, which increased in only CD8+ T cell subset (*p* = 0.01) (Figure 4A,B). Of note, while PD1 and CD38 were significantly reduced (*p* = 0.004 and *p* = 0.002, respectively), CD57 expression in CD8+ T cells increased on therapy (*p* = 0.04) (Figure 4C,D). 

### 3.3. TTV Loads and CD4+ and CD8+ T Cell Markers

Finally, considering all available PBMC samples, correlations between PBMC-associated TTV DNA loads and CD4+ and CD8+ T cell differentiation, activation, exhaustion, and senescence phenotypes were investigated. The results indicated that PBMC-associated TTV DNA loads correlated inversely with the percentage of the CD8+ CM cells (r = −0.408, *p* < 0.05) and directly with the percentages of the CD8+ EM cells and CD8+ CD57+ cells (r = 0.590, *p* < 0.01 and r = 0.464, *p* < 0.05, respectively) (Figure 5A–C).

## 4. Discussion

Evidence is progressively accumulating that TTV may represent a cheap and easy-to-measure surrogate of functional immune competence and that TTV load quantification can be especially useful in subjects with a severely immunocompromised immune system, such as in solid organ transplant recipients. In this latter context, TTV viremia is considered useful for predicting opportunistic infections and graft rejection during the post-transplant follow-up, and to establish tailor-made maintenance immunosuppression. However, other meaningful cohorts of people with different degrees of immunosuppression remain to be investigated to dissect if a continuum exists where the higher the TTV viremia, the better the correlation with the immunosuppression level. HIV infection represents an optimal model for investigating in detail the relationship between TTV and the immune system, as persons living with HIV experience different degrees of immunosuppression according to the stage of infection or the timing of ART initiation. All previous studies on TTV in HIV+ donors have been performed on chronically infected HIV+ donors and/or AIDS subjects, all displaying rather low CD4+ T cell counts [8,10,12]. Thus, the present study performed on people with acute HIV infection and with modest impairment of their immune system offers further information on this relationship and provides some interesting findings. First, it profiles the levels of TTV in PBMC of AHI and it confirms that the levels of TTV and HIV replication are highly correlated, irrespective of the type of biological samples tested (PBMC vs. plasma samples). Importantly, the inverse correlation between TTV levels and CD4+ T cell counts excluded an influence of this type of cells on viral replication. Second, our data seem not to support any predictive value of TTV on CD4+ T cell restoration after ART. This finding conflicts with previous studies carried out on chronically HIV+ donors. As already cited, Schmidt et al. [12] found higher TTV loads correlated with poorer CD4+ cell recovery after 1 year of antiretroviral therapy, while Madsen et al. [22] revealed a reduction in the plasma levels of TTV DNA after 4–5 months of ART in subjects characterized by a poor immune reconstitution. A possible explanation of the above-reported discrepancies may be due to the context in which the lymphotropic TTV was replicating, the availability of possible target cells, and the cytokine environment that may affect both TTV replication and infected cell elimination. In our cohort of AHI, we observed differences relative to HD in CD4+ and CD8+ T cells (i.e., naïve cells reduction, an increase in EM and markers of activation and exhaustion on CD8+ T cells) [23,24], and changes after 1-year of ART (i.e., a decrease in activation and exhaustion markers on CD8+ T cells), underlying the beneficial effect of antiviral therapy on the immune system [25]. Interestingly, we went further to ascertain the increase in the senescence marker on CD8+ T cells in the same time interval. This phenomenon is in line with a few reports showing the effect of different classes of antiretroviral drugs on promoting senescence in cultured cells [26,27,28]. 

Third and most important, we found correlations between TTV replication and quantitative modifications of CD4+ and CD8+ immune phenotypes. As reported above, TTV was found to correlate directly with the percentages of CD8+CD57+ T cells and CD8+ EM, while CD8+ CM are inversely associated. These latter findings suggest an enhanced TTV replication in a context where effector memory senescent CD8+ T cells predominate the scene. It is well established that CD8+CD57+ T cells clonally expand during aging, chronic infectious disease, and in some types of cancer and autoimmune disease [29]. Cytolytic CD8+ memory T cells, characterized by high expression of CD57+, display shortened telomeres and impaired proliferative responses to antigens, a condition termed immune senescence, and are enriched in subjects with chronic viral infections, such as HIV and CMV, being particularly enriched in people with HIV/CMV coinfection [30]. TTV loads have been previously associated with CD8+CD57+ T cells in autologous HSCT, where it was demonstrated that changes in CD8+CD57+ T lymphocyte expansions paralleled the kinetics of TTV viremia [31,32]. 

## 5. Conclusions

In conclusion, our study on the HIV population confirms the potential of TTV as a valuable marker of immune function, strengthening its association with the expansion of CD8+ terminal effector senescent cells in immune dysfunctions. More experiments are, however, necessary to fully address which predictive role can be played in different contexts of immunosuppression by this virus quickly evolving from an insignificant viral pathogen into a precious marker of immunity. 

## Figures and Tables

**Figure 1 viruses-15-01931-f001:**
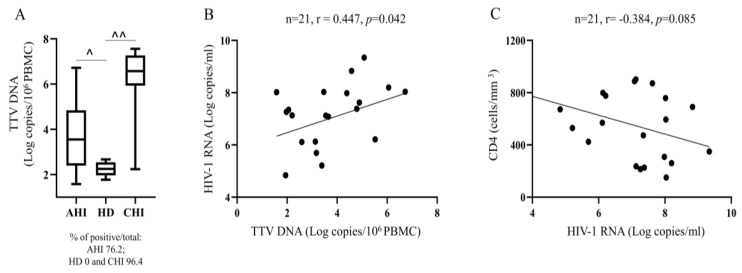
TTV DNA in AHI at serodiagnosis (T0). PBMC-associated-TTV DNA in AHI with respect to HD and CHI. (**A**) For negative TTV DNA samples, TTV loads were calculated as in Table 1, ^ *p* = 0.0041, ^^ *p* < 0.0001 in MWT; correlations between PBMC-associated TTV DNA and plasma HIV-1 RNA (**B**) and between plasma HIV-1 RNA and CD4+ T cell counts in SCT (**C**).

**Figure 2 viruses-15-01931-f002:**
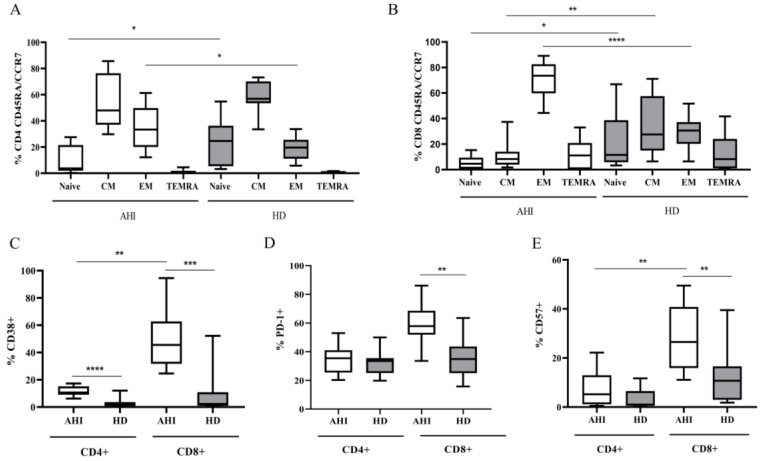
Profile of CD4+ and CD8+ T cells differentiation status (naïve cells, CM, EM, TEMRA) and markers of activation (CD38), exhaustion (PD-1), and senescence (CD57) measured in AHI at T0 and compared to HD. Differentiation profile of CD4+ T cells in AHI (light boxes), in HD (grey boxes) (**A**) and in CD8+ T cells in AHI (light boxes) and in HD (grey boxes) (**B**). Frequency of CD38+ CD4+ and CD8+ T cells in AHI (light boxes) and HD (grey boxes) (**C**). Frequency of PD1+ CD4+ and CD8+ T cells in AHI (light boxes) and HD (grey boxes) (**D**). Frequency of CD57+ CD4+ and CD8+ T cells in AHI (light boxes) and HD (grey boxes) (**E**). Asterisks indicate the statistical significance of the comparisons of the median percentages between AHI and HD: * *p* = 0.01 in naïve CD4+ and CD8+; * *p* = 0.02 in EM CD4+; ** *p* = 0.003 in CM CD8+; **** *p* = 0.0001 in EM CD8+; *** *p* = 0.003 in CD38 CD8+; **** *p* = 0.0001 CD38 CD4+; ** *p* = 0.004 CD57 CD8+; ** *p* = 0.003 in PD-1 CD8+. In AHI: ** *p* = 0.002 in CD38 CD8+, CD4+, and in CD57 CD8+. WT and MWT were used.

**Figure 3 viruses-15-01931-f003:**
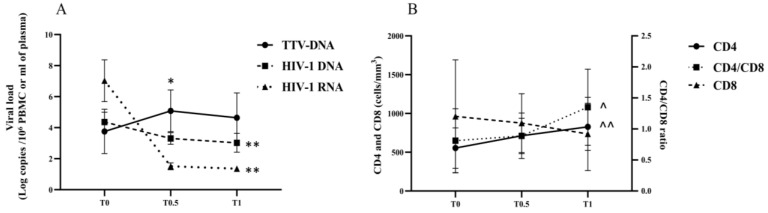
Kinetics of viro-immunological parameters in AHI subjects during 1 year of ART. TTV DNA, HIV-1 DNA, and HIV-1 RNA viral load at T0, T0.5 (3 months of ART), and T1 (1 year of ART) (**A**). CD4+ and CD8+ cell counts and CD4/CD8 ratio at T0, at T0.5 (3 months of ART), and at T1 (1 year of ART) (**B**). The significances (by MWT) of the statistical comparisons between median values at the different times points with respect to T0 values are shown. (**A**): * *p* = 0.02, ** *p* < 0.0001; (**B**): ^ *p* = 0.008, ^^ *p* = 0.009.

**Figure 4 viruses-15-01931-f004:**
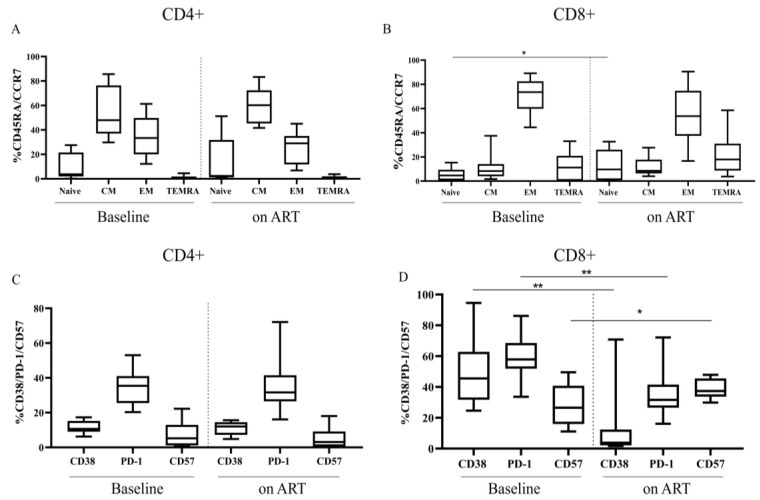
CD4+ and CD8+ phenotype comparison between AHI at baseline and under antiretroviral therapy. Frequency of CD4+ and CD8+ differentiation phenotypes in AHI at baseline T0 (**A**) and on therapy T0.5 + T1 (**B**). Frequency of CD4+ and CD8+ T cells displaying the expression of the activation (CD38), the exhaustion (PD-1), and the senescence marker (CD57) at baseline T0 (**C**) and on therapy T0.5 + T1 (**D**). Asterisks indicate the statistical significance of the comparisons of the median percentages between AHI at baseline and on ART: * *p* = 0.01 in naïve CD8+; ** *p* = 0.002 in CD38 CD8+; ** *p* = 0.004 PD-1 CD8+; * *p* = 0.04 CD57 CD8+; MWT was used.

**Figure 5 viruses-15-01931-f005:**
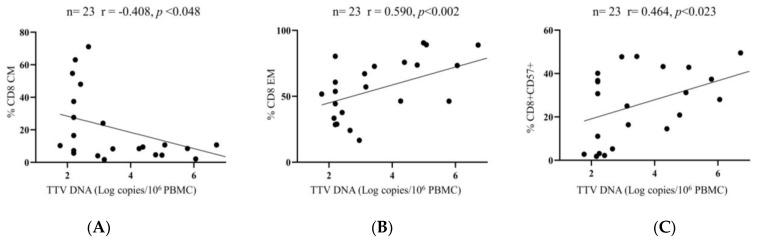
Correlation between TTV loads and percentage of CD8 CM, CD8 EM, and CD8+CD57+ cells. TTV loads were inversely correlated (by SCT) with the frequencies of CD8+ CM (**A**), while they were directly correlated with both the frequencies of CD8+ EM (**B**) and the frequencies of CD8+CD57+ cells (**C**).

**Table 1 viruses-15-01931-t001:** Viro-immunological features of AHI.

Hiv+ Donors	Sex	Age (yrs)	Fiebig Stage	CD4 (cells/mm^3^)	CD4/CD8 Ratio	HIV-1 RNA (Log Copies/mL)	HIV DNA (Log Copies/ 10^6^ PBMC)	TTV DNA (Log Copies/ 10^6^ PBMC)
1	M	38	II/III	594	0.3	8.0	3.6	3.5
2	M	47	II/III	871	0.7	7.6	4.1	4.9
3	M	54	II/III	529	1.0	5.2	4.3	3.4
4	M	45	II/III	887	1.5	7.1	5.0	3.6
5	M	71	II/III	569	1.6	6.1	4.8	2.6
6	M	31	II/III	776	2.1	6.2	3.7	5.5
7	M	46	II/III	901	1.1	7.1	4.0	3.5
8	F	41	IV	308	0.4	7.9	4.8	4.4
9	M	39	IV	672	0.6	4.8	3.7	1.9 ^
10	M	60	IV	214	0.1	7.3	4.1	1.9 ^
11	M	54	II/III	349	1.5	9.3	4.7	5.1
12	M	35	II/III	690	0.2	8.8	5.4	4.6
13	M	40	IV	936	1.0	3.9	2.9	4.0
14	M	30	II/III	473	0.6	7.3	4.1	2.1 ^
15	M	27	II/III	758	0.6	8.0	5.1	1.6 ^
16	M	50	II/III	236	0.8	7.1	5.2	2.2 ^
17	M	36	II/III	150	0.7	8.0	4.2	6.7
18	M	33	IV	798	0.8	6.1	3.9	3.1
19	M	32	IV	424	0.6	5.7	4.5	3.2
20	M	23	IV	226	0.8	7.4	5.1	4.8
21	M	33	II/III	260	0.2	8.2	4.2	6.1

^ TTV loads calculated by the following formula: [LOD value (3 copies per reaction) × 10^6^ cells]/no. cells in the sample.

## Data Availability

Data are contained within the article, additional raw data may be available upon request.

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
