# Peer review of "Kinetics of TTV Loads in Peripheral Blood Mononuclear Cells of Early Treated Acute HIV Infections"

_viruses, 2023, doi:10.3390/v15091931_

Round 1
Reviewer 1 Report
Abbate et al. Kinetics of TTV loads in peripheral blood mononuclear cells of early treated acute HIV infections. This well-written manuscript provides further information regarding TTV load in people living with HIV treated in acute infection. Several clarifications/ modification would help clarify the manuscript as well as further statistical testing could be performed in order to provide a solid ground to discuss results from previous studies. Please, find below my comments
General
Please, change “patients” for “people living with HIV (PLWH)” or “HIV+ donors” throughout the paper
Indicate the statistical test used in each figure. I also suggest to show the actual p value rather than p<0.05…etc….throughout the manuscript
How did individuals with higher CD4 T cell counts (i.e., split by the median) compared to those with lower CD4 counts? Those would be helpful to discuss the results in the context of previous studies performed in individuals with low counts. In particular, may authors clarify lines 260-262? Have authors run a multiparametric test to identify factors independently associated with TTV viral load and CD4 counts? These analysis could strengthen the conclusions, especially when discussing the potential predictive value of TTV (lines 263-265)
Major
Where there any correlations between HIV DNA and other parameters including HIV RNA, TTV RNA/DNA/CD4?
Lines 163-166, I would recommend to show graphs comparing the three groups as part of figure 1. In fact, these should probably be Figure 1A, 1B…etc…and then the correlations should go next.
Table 2 is not clearly explained. What is the meaning of 16/21…etc..? What is the goal of this table? Please, clarify in the text.
Line 177: I would suggest to name the phenotype naïve/memory rather than differentiation
Resolution and size of the figures must be increased. It was very hard to review the printed form of the manuscript
Line 204: a p value p=0.05 is not statistically significant.
Line 208: unclear the relation authors are trying to show “percentages of patients”? Where is the data?
Line 211: Is on-ART after one year? Please, clarify.
Please, correct CD45 to CD45RA to define naïve/memory populations in figure 4. Where frequencies of naïve/memory populations comparable to HD after one year on ART? An additional figure or supplementary figure could be added.
Minor
Line 77: Spell out INMI
Line 85: Specify whether these 28 participants were on ART and type of ART
Line 190: Correct the duplicated information in the figure legend
Line 222: Correct the grammar
Minor editing
Author Response
Review 1.
Comments and Suggestions for Authors
General.
Please, change “patients” for “people living with HIV (PLWH)” or “HIV+ donors” throughout the paper
Reply: the change has been done.
Indicate the statistical test used in each figure. I also suggest to show the actual p value rather than p<0.05…etc….throughout the manuscript
Reply: the statistical test used in each figure has been indicated and the p values have been shown.
How did individuals with higher CD4 T cell counts (i.e., split by the median) compared to those with lower CD4 counts? Those would be helpful to discuss the results in the context of previous studies performed in individuals with low counts. In particular, may authors clarify lines 260-262? Have authors run a multiparametric test to identify factors independently associated with TTV viral load and CD4 counts? This analysis could strengthen the conclusions, especially when discussing the potential predictive value of TTV (lines 263-265)
Reply: As suggested by the reviewer, a multivariable logistic regression analysis has been performed using CD4 T cell counts categorized by median value as ≥ or<150 cells/mm3 as dependent variable and TTV DNA, HIV-1 RNA and HIV-1 DNA values as independent variables. Results indicated that TTV load was the only variable independently associated with CD4 T cell counts, increasing in values when CD4 T cell counts decreased [p=0.003, O.R. 0.51 (95% C.I. 0.33-0.79]. A sentence with these additional results has been added. Again, the sentence in lines 298-300 has been clarified.
Major
Where there any correlations between HIV DNA and other parameters including HIV RNA, TTV RNA/DNA/CD4?
Reply: HIV-1 DNA did not correlate with either HIV-1 RNA, TTV DNA or CD4 T cell counts at T0.
Lines 163-166, I would recommend to show graphs comparing the three groups as part of figure 1. In fact, these should probably be Figure 1A, 1B…etc…and then the correlations should go next.
Reply: data comparing TTV DNA among the three groups of subjects have been included as part of Figure 1 (panel A).
Table 2 is not clearly explained. What is the meaning of 16/21…etc..? What is the goal of this table? Please, clarify in the text.
Reply: Table 2 is now replaced by panel A in Figure 1 describing the comparison between TTV DNA load among the subjects’ groups. In some cases, we found TTV DNA negative samples and the ratio 16/21 meant positive/total samples tested and this is the same for 0/5 and 27/28.
Line 177: I would suggest to name the phenotype naïve/memory rather than differentiation
Reply: according to your suggestion, we replaced phenotypic profile rather than differentiation, since we analyzed also effector memory subgroups, see line 186.
Resolution and size of the figures must be increased. It was very hard to review the printed form of the manuscript
Reply: the resolution of the figures has been increased to 600dpi.
Line 204: a p value p=0.05 is not statistically significant.
Reply: we amended the phrase according to your suggestion, see line 220.
Line 208: unclear the relation authors are trying to show “percentages of patients”? Where is the data?
Reply: Ok, we clarified the sentence see lines 223-228.
Line 211: Is on-ART after one year? Please, clarify.
Reply: AHI subjects were ART treated as soon as HIV serodiagnosis was performed and were on-ART after one year. We better clarified this point see lines 217-218.
Please, correct CD45 to CD45RA to define naïve/memory populations in figure 4. Where frequencies of naïve/memory populations comparable to HD after one year on ART? An additional figure or supplementary figure could be added.
Reply: we amended the figure 4 accordingly with the reviewer’s suggestion.
Concerning the question if the frequencies of naïve/memory populations were comparable to HD after one year on ART, we have very few number of observations (only 3) and we think that any conclusions on these few data could be not relevant for the paper.
Minor
Line 77: Spell out INMI
Reply: done
Line 85: Specify whether these 28 participants were on ART and type of ART
Reply: at the moment of the evaluation, the CHI were naïve to treatment. We clarified this point see line 87.
Line 190: Correct the duplicated information in the figure legend
Reply: done
Line 222: Correct the grammar
Reply: done
Reviewer 2 Report
Congratulations. This is an excellent study. Two minor issue:
Any known direct effect of ART on TTV replication?
Please rephrase first sentence in the Abstract's Results section.
Author Response
Review 2
Comments and Suggestions for Authors
Congratulations. This is an excellent study. Two minor issue:
Any known direct effect of ART on TTV replication?
Reply: to date, except for IFN, no anti-viral drug has been demonstrated to directly act on TTV replication.
Please rephrase first sentence in the Abstract's Results section.
Reply: as suggested, the first sentence has been rephrased.